# Segment Vasculature in 3D Scans of Human Kidney

**Ruitao Jing**
2024312583
Shenzhen International Graduate School
Tsinghua University
jingrt24@mails.tsinghua.edu.cn

**Lily Sheng**
2024280034
Department of Computer Science
Tsinghua University
cqr24@mails.tsinghua.edu.cn

**Jessica Zhou**
2024400452
Department of Computer Science
Tsinghua University
cy-zhou24@mails.tsinghua.cn

## 1 Background

Vasculature Common Coordinate Framework (VCCF) aims to create a comprehensive map of blood vessels in the human body to help researchers understand cellular interaction and functions. However, VCCF faces data gaps due to the extensive manual effort required for annotating and segmenting vascular structures, with each dataset taking up to six months to complete. Machine learning approaches currently struggle to generalize across new datasets due to anatomical variability and varying image quality from HiP-CT advancements. Hierarchical Phase-Contrast Tomography (HiP-CT) is an advanced imaging technique to produce 3D multi-resolution imaging datasets, capturing micron-level details to entire intact organs. HiP-CT is crucial for accurate segmentation and mapping.

In this project, we will develop a new machine learning model to automatically segment blood vessels using HiP-CT data from human kidneys (Figure 1). Our goal is to improve the VCCF by providing accurate and detailed mapping of vasculature.

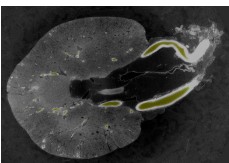

Figure 1: The segmentation example of human kidney

## 2 Definition

The problem can be formally defined as a 3D segmentation task. Given a high-resolution 3D image of a kidney, we aim to generate a corresponding 3D segmentation mask that accurately identifies and isolates the kidney's blood vasculature within each slice. This involves pixel-wise classification, where each pixel (or voxel, in 3D) in the image is labeled as either part of the vasculature or not.

## 3 Related Works

Researchers are working on developing VCCF using known vascular pathways as axes to represent the relative positions of cells and other anatomical elements using imaging methods like HiP-CT [13][5][17][12]. The rise of convolutional neural networks (CNNs) and deep learning has introduced new image segmentation methods. U-Net, a widely used architecture for segmentation, comprises of a contracting network with pooling layers and an upsampling network. For localization, the results from the upsampled output are combined with the results from the high resolution features from the contracting path [10][19]. Such architecture have been successfully applied in medical imaging

and 3D image segmentation [8][4][20]. Some works integrate attention with U-Net to highlight regions that require more focus [9][11][7][2]. Other works utilize residual blocks from ResNet in the U-Net encoder to improve feature extraction and gradient flow [3][6]. Researchers have also applied Diffusion Probabilistic Model (DPM) and denoising methods with U-Net to achieve higher segmentation accuracy in images that have noise and uncertainty [18][14][16][15].

# 4  Proposed Method

## 4.1  Motivation and Baseline

Our project is based on a finished Kaggle competition. After reviewing solutions of the top-performing teams, we selected the second-place approach as our baseline. The primary reason for this choice is that the solution employs a relatively standard U-Net3D model without relying on complex or unconventional tricks. This straightforward yet effective design addresses the task efficiently and demonstrates strong performance, making it a solid foundation for further development.

The dataset provided by the organizers contains a significant number of sparsely labeled images, which limits the potential for further performance improvement through supervised learning alone. Therefore, we aim to enhance the model's generalization ability by leveraging effective pretraining techniques. Inspired by [1], we will apply denoising pretraining to the decoder part of the baseline U-Net 3D model while frozen the encoder part. Afterward, the entire model is fine-tuned on a densely annotated dataset to achieve better performance.

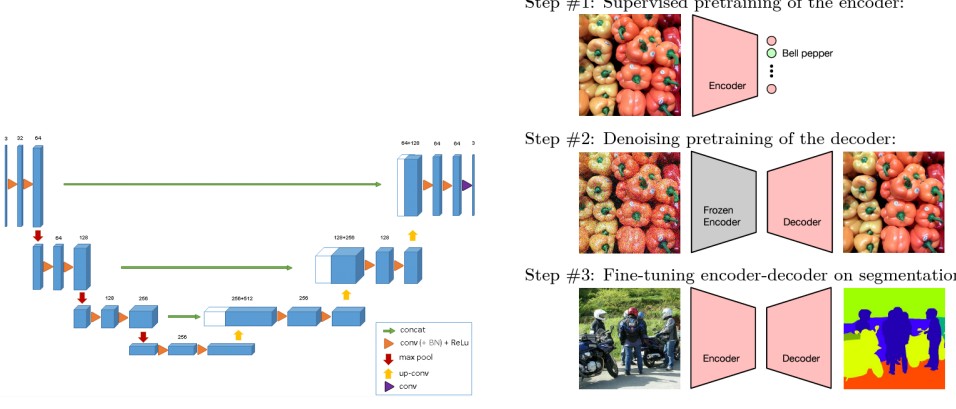

(a) Baseline: The framework of U-Net3D from [20]    (b) Improvement: DDeP techniques from [1]

Figure 2: The pipeline of our work

## 4.2  Dataset

This dataset from Kaggle comprises high-resolution 3D images of several kidneys with 3D segmentation masks of their vasculature.

The training dataset contains TIFF scans, representing 2D slices extracted from a 3D volume, sequentially aligned along the z-axis. Accurate analysis requires stacking these slices depth-wise. The dataset is further subdivided into several distinct subsets. Subsets $kidney_{1dense}$ and $kidney_{3dense}$ are densely annotated while $kidney_2$ and $kidney_{3sparse}$ are sparsely processed.

The testing dataset includes subsets $kidney_5$ and subset $kidney_6$. These may differ in beamline or resolution from those used in the training set. More details about the dataset can be seen in appendix.

## 4.3  Implementation

Our project is divided into two main steps (Shown in Figure 2). First, we will reproduce the approach of the second-place solution mentioned in section 4.1. Next, we are going to improve the model based on decoder DDeP techniques[1] and continuously optimized the model to enhance its performance. The optimized models will be submitted to Kaggle's Late Submission for evaluation, and the best-performing model will be selected as the final model for our report.

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

# 5   Supplementary Material

## 5.1   Explanation of the Dataset

The dataset provided by the Kaggle competition consists of high-resolution 3D kidney images accompanied by 3D segmentation masks of their vasculature. The objective is to generate segmentation masks for the test set kidney datasets. The images were captured using Hierarchical Phase-Contrast Tomography (HiP-CT), a technique that provides detailed 3D imaging of ex vivo organs with resolutions ranging from 1.4 μm to 50 μm.

The training data is organized into TIFF scans, with each scan representing a 2D slice of a 3D volume, sequentially aligned along the z-axis. Corresponding blood vessel segmentation masks are provided in separate TIFF files. The following subsets are included in the dataset:

- $kidney_{1dense}$: A right kidney imaged at 50 μm resolution (BM05) with dense segmentation of the vascular tree down to two generations beyond the glomeruli.

- $kidney_{1voi}$: A high-resolution (5.2 μm) subset of $kidney_1$.

- $kidney_2$: A kidney from another donor at 50 μm resolution, with 65% sparse segmentation.

- $kidney_{3dense}$: A 500-slice portion at 50.16 μm resolution (BM05) with dense segmentation (only labels are provided).

- $kidney_{3sparse}$: The remaining sparse masks (85%) for $kidney_3$, with images stored in the $kidney_{3sparse}$ folder.

The test set ($kidney_5$, $kidney_6$) contains TIFF scans, which may vary in beamline or resolution from the training set.

A train_rles.csv file provides run-length encoded masks, with each slice identified by a {dataset}_{slice} format.

Additionally, public and private test datasets include continuous 3D kidney sections scanned using HiP-CT. The public test set is binned to 50.28 μm/voxel (from 25.14 μm/voxel), while the private set is binned to 63.08 μm/voxel (from 15.77 μm/voxel). These datasets are distinct from those in the training set, ensuring independent evaluation. The full test set contains about 1500 TIFF images.

The complete dataset contains 14,365 files and occupies 43.52 GB.

