# OpenReview forum: "[Proposal-ML] Segment Vasculature in 3D Scans of Human Kidney"
_tsinghua.edu.cn/THU/2024/Fall/AML — THU 2024 Fall AML Submission_

### Official Review · ~Chan_Thong_Fong1 · 2024-11-10
**Practical approach on automating segmentation of vasculature in 3D kidney scans**

**Rating:** 9
**Confidence:** 4

**Review:**

This paper provides a thorough analysis of automating the segmentation of vasculature in 3D kidney scans using advanced machine learning methods. The authors address critical challenges in the field, such as the intensive manual work required for annotating vascular structures and the variability inherent in anatomical data, which complicates generalization across datasets. They build on existing architectures like U-Net3D, incorporating denoising pretraining techniques to enhance segmentation performance. The structure of the paper is clear, detailing the motivation behind the work, the dataset used, and the step-by-step methodology, which includes reproducing and refining an effective solution from a Kaggle competition. The proposed approach shows potential for advancing the creation of a more detailed Vasculature Common Coordinate Framework (VCCF), offering valuable insights for research and clinical applications that depend on accurate vascular maps. This work is a meaningful addition to the field and demonstrates a solid grasp of current challenges and innovative solutions.

---

### Official Review · ~Rim_El_Filali1 · 2024-11-11
**Good Proposal but Lacking Clear Evaluation Metrics**

**Rating:** 9
**Confidence:** 4

**Review:**

This proposal addresses the challenge of automating the segmentation of blood vessels in the human kidney using 3D HiP-CT scans. The aim is to contribute to the development of the CCF by enhancing the precision and efficiency of vasculature mapping in kidney scans.

Pros:
- The use of a top-performing Kaggle solution as a baseline ensures that the approach is rooted in successful strategies.
- The goal of improving VCCF is highly relevant to advancing the understanding of human anatomy and aiding medical research.

Cons:
- The proposal does not provide a clear explanation of how the new optimizations might outperform existing solutions.
- The proposal briefly mentions the use of Kaggle’s Late Submission for evaluation but does not provide specific details about how the performance of the final model will be comprehensively assessed beyond Kaggle scores. More explicit metrics for evaluating segmentation accuracy and robustness across different kidney types would strengthen the evaluation plan.

---

### Official Review · ~Nan_Sun10 · 2024-11-11
**An Ambitious Step Forward in 3D Kidney Vasculature Segmentation, But Practical Challenges Remain**

**Rating:** 8
**Confidence:** 3

**Review:**

This proposal presents a machine learning approach to segment blood vasculature in 3D kidney scans using HiP-CT data, addressing key limitations in the Vasculature Common Coordinate Framework (VCCF). By leveraging a U-Net 3D baseline model and introducing denoising pretraining for the decoder, the project aims to enhance segmentation accuracy in sparsely annotated datasets, which could accelerate the generation of high-resolution vascular maps for medical research.

While the concept is promising, several challenges merit attention. First, the dependence on high-quality 3D imaging data, such as HiP-CT, may limit the generalizability and scalability of this approach to other imaging techniques and organs. Additionally, while the use of DDeP is innovative, it lacks a comprehensive plan for addressing dataset-specific variances, such as resolution and anatomical variability, which are critical for clinical applications.

---

### Official Review · ~Zhang_Mingkang1 · 2024-11-11
**Interesting**

**Rating:** 9
**Confidence:** 3

**Review:**

Strengths:

Background


Clear scientific impact for VCCF development.
Well-defined medical imaging context.
Strong practical motivation for automated segmentation.
Excellent connection to current challenges in the field.


Definition


Clear formulation of 3D segmentation task.
Well-defined problem scope.
Good explanation of data characteristics.
Clear connection between input and desired output.


Related Work


Comprehensive review of U-Net variations.
Excellent coverage of attention mechanisms.
Strong analysis of current limitations.
Well-organized literature survey covering both traditional and modern approaches.


Proposed Method


Clear baseline selection with justification.
Innovative use of denoising pretraining.
Well-structured implementation plan.
Detailed dataset analysis and handling.

Key Comments:

Strong technical foundation with U-Net3D baseline.
Innovative improvement using decoder DDeP techniques.
Well-thought-out dataset handling strategy.
Clear experimental validation plan.
Excellent supplementary materials with detailed dataset explanation.

Areas for Improvement:

Could provide more details on the denoising pretraining process.
More specific metrics for evaluating segmentation quality.
Consider adding ablation studies for different components.
Could elaborate on computational requirements.


This is a well-structured proposal with clear scientific merit, strong methodology, and practical medical applications.

---

### Official Review · ~Xin_Chen65 · 2024-11-11
**An interesting approach**

**Rating:** 8
**Confidence:** 3

**Review:**

The proposal presents a well-structured and focused project aimed at advancing the Vasculature Common Coordinate Framework (VCCF) by developing a machine learning model to automatically segment blood vessels in 3D scans of human kidneys. Overall, the proposal is well-written and presents a clear plan for addressing a significant challenge in medical imaging. With a few additional details and clarifications, it could be even stronger.

strength: (1) The problem is clearly defined as a 3D segmentation task, with the goal of generating accurate 3D segmentation masks for kidney vasculature. The pixel-wise classification approach is well articulated. (2) The motivation for choosing the baseline model is clear and logical. The proposal to use denoising pretraining to enhance the model's generalization ability is innovative and addresses the limitations of the dataset. (3) The dataset description is detailed, providing a clear understanding of the data subsets and their characteristics, which is essential for potential users or reviewers to grasp the scope of the project.

weakness: (1) While the proposal outlines a solid approach, it could benefit from a clearer statement on the novelty of the proposed method compared to existing solutions. What makes this approach unique or superior? (2) Will there be specific metrics used to measure accuracy, such as Dice coefficient, IoU (Intersection over Union), or others?

---

### Official Review · ~Zihan_Lv1 · 2024-11-11
**Good topic, looking forward to your result**

**Rating:** 9
**Confidence:** 4

**Review:**

The selected topic is based on the mature U-Net 3D architecture as baseline, and introduces DDeP for improvement, which is a novel idea with clear research value. However, the project is mainly a combined application of existing methods, and it is suggested to propose more innovative points in model architecture or training strategy.

---

### Official Review · ~Killian_Conyngham1 · 2024-11-12
**Review of Segment Vasculature in 3D Scans of Human Kidney**

**Rating:** 10
**Confidence:** 4

**Review:**

This is an excellent proposal with a clearly engaging premise. In the Background section, the problem and its relevance is introduced with clear detail. The related works section provides a good overview of the literature and clear evidence of the potential for the proposed approach. The Motivation and Baseline and Dataset sections clearly outlines the Kaggle dataset being used, and why the second place submission is being used as a baseline for the approach. The use of relevant and clear visual aids is especially appreciated. The main suggestion for improvement would be to add detail to the Implementation section, going specifically into more detail into how exactly the authors propose to improve their baseline model and regarding what evaluation criteria the Kaggle competition uses.

---

### Official Review · ~Ruilin_Hu2 · 2024-11-12
**Review of Segment Vasculature in 3D Scans of Human Kidney**

**Rating:** 9
**Confidence:** 5

**Review:**

The proposal for segmenting vasculature in 3D kidney scans presents a well-defined approach to improve the Vasculature Common Coordinate Framework (VCCF).

Strengths:
	1.	The use of Hierarchical Phase-Contrast Tomography (HiP-CT) for high-resolution imaging is innovative and offers promising accuracy.
	2.	The combination of a straightforward U-Net3D model with denoising pretraining (DDeP) enhances model generalization and segmentation precision.
	3.	The structured dataset and planned late Kaggle submissions show a well-organized implementation strategy.

Weaknesses:
The reliance on sparse annotations may limit the segmentation’s accuracy, particularly for complex structures.


Overall, the project has significant potential to contribute to vascular mapping advancements.

---

### Official Review · ~Zhaoxi_Li2 · 2024-11-12
**Review of Segment Vasculature in 3D Scans of Human Kidney**

**Rating:** 9
**Confidence:** 4

**Review:**

This proposal presents a strong approach to automating 3D segmentation of kidney vasculature, addressing critical challenges in the Vasculature Common Coordinate Framework (VCCF) by combining advanced HiP-CT imaging with a U-Net3D model. The proposal is well-motivated and methodologically sound, with a clear plan to tackle limitations in manual annotation through denoising pretraining to improve segmentation accuracy. The inclusion of diverse dataset types strengthens the evaluation strategy, allowing for robust testing across various image qualities and segmentation densities. However, more detailed plans for evaluation metrics and an outline of comparative and ablation studies would enhance the proposal's rigor and clarity. Overall, the proposal is highly promising, with the potential for impactful contributions to vascular segmentation and medical imaging.

---

### Official Review · ~Bowen_Gao1 · 2024-11-12
**Review of Segment Vasculature in 3D Scans of Human Kidney**

**Rating:** 8
**Confidence:** 4

**Review:**

## Summary
This proposal addresses a Kaggle competition focused on the segmentation of human kidney images. The authors propose to use the DDeP technique to enhance the performance of the competition’s second-best model.

## Strengths

1. The problem is clearly defined and easy to understand.
2. Related work is thoroughly reviewed, providing a solid background.
3. Datasets and methods are well-documented.

## Weaknesses

1. The figures illustrating the framework are directly taken from other papers; creating original diagrams would be better.
2. The level of novelty in the approach is somewhat limited.